# OpenReview forum: "BackdoorDM: A Comprehensive Benchmark for Backdoor Learning on Diffusion Model"
_NeurIPS.cc/2025/Datasets_and_Benchmarks_Track — NeurIPS 2025 Datasets and Benchmarks Track poster_

### Official Review · Reviewer_Yt6A · 2025-06-24

**Rating:** 4
**Confidence:** 3

**Summary:**

This paper presents BackdoorDM, the first comprehensive benchmark for backdoor learning on diffusion models. It incorporates nine state-of-the-art attack methods, four defense techniques, and three visualization tools, along with a unified evaluation framework based on multimodal large language models (MLLMs). The benchmark categorizes backdoor attacks and target types in diffusion models, introduces standardized evaluation metrics (e.g., ASR, PSR, ACC), and provides detailed comparisons across multiple datasets and diffusion architectures.

**Additional Feedback:**

None

**Dataset Code Accessibility:**

Yes

**Dataset Code Comments:**

The authors provide code and standardized pipelines for evaluating several attack/defense methods.

**Ethical Considerations:**

No, there are no or only very minor ethics concerns

**Final Justification:**

Most of my concerns are addressed. I would increase my rating to borderline accept.

**Limitations Weaknesses:**

1. Narrow scope of covered domains: Despite the “comprehensive” claim, the benchmark is limited to unconditional and text-to-image diffusion models. It does not cover other crucial modalities such as text-to-speech (TTS) or text-to-video (T2V), even though they are discussed briefly. The current version thus falls short of fully generalizing across the broader generative model landscape.

2. Limited novelty in methodology: The core contributions are primarily engineering and integration efforts. Most attack/defense techniques and evaluation metrics are adopted from existing work rather than innovated upon.

3. Too few original references for foundational attacks: Many attack methods implemented are sourced from a small group of recent papers (e.g., RickRolling, BadT2I, EvilEdit), and the diversity in design paradigms is still limited. This makes the benchmark susceptible to overfitting to these methods rather than generalizing broadly.

4. Evaluation still partly dependent on qualitative judgment: While MLLM-based evaluation offers some automation, it introduces ambiguity in consistency and reproducibility compared to classifier-based metrics. For example, GPT-based PSR and ACC may fluctuate depending on prompt phrasing or model variants.

5. No analysis of real-world usability or robustness: The benchmark lacks discussion on practical constraints (e.g., runtime trade-offs, defense transferability across architectures, or zero-day attacks). It also does not assess robustness under adversarial adaptations or adaptive attacks.

**Strengths Contributions:**

1. The paper addresses a timely and relevant problem — the lack of standardized benchmarks for evaluating backdoor attacks on diffusion models, which are becoming increasingly prevalent in AIGC systems.

2. It provides a structured taxonomy of attack and target types that are unique to generative diffusion models, highlighting key distinctions from backdoor learning in discriminative settings.

3. The integration of MLLMs for unified evaluation (ASR, PSR, ACC) is novel and offers a practical solution to the fragmented and inconsistent evaluation protocols in prior works.

4. The benchmark covers a wide range of attack methods and models, including unconditional DMs and text-to-image generation pipelines (e.g., Stable Diffusion v1.5 and v2.0).

5. Visualization tools (e.g., activation norms and attention maps) offer some interpretability for model behavior under attack.

---

> ### Author Rebuttal · Authors · 2025-07-31
>
> Thank you very much for your careful review of our paper and thoughtful comments. We are greatly encouraged by your positive feedback on our **timely benchmark**, **structured taxonomy**, **novel unified evaluation**, **wide range implementation**, and **interpretable visualization**. We hope the following responses could help alleviate your concerns.
>
> ---
>
> ### **Cons1: Concern about the generalizability to TTS and T2V.**
>
> **R1:** Thanks for your concern. We acknowledge the importance of TTS and T2V paradigms, and have discussed the limitations in the Conclusion Section. As the first benchmark on diffusion backdoor, our main goal is to unify the mainstream paradigms and fairly compare the current methods to inspire future work. We achieved this goal by reproducing the ‘comprehensive’ methods on the current backdoor research focus, i.e., unconditional and text-to-image (T2I) diffusion models, into one comparable framework and proposing a unified MLLM evaluation method. As far as we know, by the submission date of NeurIPS’25, we have covered most of the methods in this field, belonging to unconditional and T2I DM. In contrast, TTS and T2V paradigms are still far behind, with only one recent Arxiv work on T2V, making them incomparable and out of scope in our framework. As our evaluation framework and codebase are extensible to those paradigms with only slight modifications, we will update them when more works emerge.
>
> ---
>
> ### **Cons2: Concern about the novelty and contribution.**
>
> **R2:** Thanks for your concern about the novelty and contribution. Except for the engineering effort based on the existing works, our contributions also lay in several innovative aspects.
>
> - **New unified formulation**. We introduce Equation (1) to unify the current backdoor injection methods and offer a comprehensive taxonomy on attack types and target types. The formulation can also inspire future research, where we propose a new target type, ObjectAdd, in Section 3.2 based on it as an example.
> - **New unified evaluation.** We introduce MLLM evaluation to provide a unified assessment of both model specificity and utility across all target types, where the previous methods suffer from training different classifiers for the diverse targets. We also propose a new metric, PSR, to evaluate the non-backdoor target, which is essential but overlooked before.
> - **New visualization tools**. We adapt three useful visualization tools to observe the behaviors of backdoored DM and inspire more potential defenses, where two of them have not yet been used in the field of diffusion backdoor.
> - **New findings.** We reveal several important findings from our experimental results to inspire future research in this field. We would like to summarize them as follows for your reference:
>
>     > Our extensive experiments yield several important findings, encompassing the results of current attacks, defenses, and the visualization of backdoored models.
>     > - **Attacks**: 1) Among the existing backdoor targets, ImageFix and StyleAdd achieve higher ASRs, indicating a higher threat to society. 2) The ObjectRep target attracts the most research attention; however, the current attacks fail to balance well across different criteria. 3) Non-conflicting backdoor targets may be an easier task to attack.
>     > - **Defenses**: The current performances of model-level defenses are limited on both unconditional DM (Elijah) and T2I DM (T2IShield and Textual Perturbation), whereas the input-level method, TERD, is effective.
>     > - **Visualizations**: 1) *Assimilation Phenomenon*: The ImageFix target exhibits a more obvious phenomenon, while other target types may fail, indicating the limitation of using the attention map to observe backdoors. 2) *Activation Norm*: Certain neurons in backdoored DMs consistently exhibit significant activation norm differences between clean and poisoned samples, inspiring the potential defense for identifying backdoored neurons. 3) *Pre-Activation Distribution*: A clear bimodal pattern can be observed in certain neurons, inspiring more future discussions on neuron functionality for defense.
>     >
>
> ---
>
> ### **Cons3: Concern about the generalizability to different attack paradigms.**
>
> **R3:** Thanks for your concern. As an emerging field since 2023 [1,2], our benchmark already covers most of the existing methods and aligns with their design paradigms, as you also recognize in Strengths 4 above. To address the generalizability challenge across different design paradigms, we introduce Equation (1) to unify the methods we have implemented and to inspire future research, exemplified by a novel attack method, ObjectAdd. Our codebase is also designed in a modular fashion, making it easy to implement and integrate new attack paradigms into the framework. Therefore, we believe the benchmark is sufficiently generalizable. We will continue to track the latest attack methods to further enrich our benchmark, and we hope our work contributes to a more trustworthy AIGC community.
>
> ---
>
> ### **Cons4: Concern about the consistency of MLLM evaluation.**
>
> **R4:** We appreciate your concern regarding the consistency and reproducibility of MLLM evaluation. Similar to other T2I evaluation studies [3,4], this potential consistency issue is recognized as a common challenge in the field. On the one hand, we have implemented traditional evaluation metrics from previous works, as described in Section 3.4, to ensure the usability of our framework without relying on MLLMs. On the other hand, to ensure the reproducibility of our MLLM-based evaluation results, we provide the MLLM prompts in Appendix B.7, the specific MLLM versions in Appendix C.1, and the experimental settings in Appendix B.6.
>
> Here, we further evaluate the performance consistency of MLLM-based evaluations (using GPT-4o and Qwen2-VL-72B) across three random seeds: 0, 100, and 678. The results are presented below (Tab. 1). We observe that most results range within 1%, indicating high consistency when using either closed-source (GPT-4o) or open-source (Qwen2.5-VL) MLLMs. Therefore, the MLLM-based evaluation results can be considered trustworthy, as long as users run the code under our released settings.
>
> **Tab. 1. MLLM evaluation results comparison on different random seeds. (%)**
>
> |ObjectRep|MLLM|GPT-4o|GPT-4o|GPT-4o|GPT-4o|Qwen2.5-VL-72b|Qwen2.5-VL-72b|Qwen2.5-VL-72b|Qwen2.5-VL-72b|
> -|-|-|-|-|-|-|-|-|-
> Attacks|Random Seed|0|100|678|**Range**|0|100|678|**Range**
> TPA (RickRolling)|ACC|82.61|83.42|83.41|**0.81**|78.29|78.17|78.33|**0.16**
> TPA (RickRolling)|ASR|96.80|96.80|96.80|**0.00**|96.90|96.90|96.90|**0.00**
> TPA (RickRolling)|PSR|5.01|4.43|5.50|**1.07**|2.86|3.16|3.22|**0.36**
> DB (Paas)|ACC|70.89|71.29|70.87|**0.42**|64.70|64.51|64.70|**0.19**
> DB (Paas)|ASR|51.40|51.20|51.30|**0.20**|51.45|51.50|51.70|**0.25**
> DB (Paas)|PSR|60.45|59.64|60.22|**0.81**|52.06|52.29|52.02|**0.27**
> EvilEdit|ACC|83.82|83.83|83.01|**0.82**|79.62|79.85|79.46|**0.39**
> EvilEdit|ASR|61.00|60.60|61.10|**0.50**|59.50|60.20|59.70|**0.70**
> EvilEdit|PSR|84.30|83.89|85.25|**1.36**|75.94|76.52|75.55|**0.97**
>
> ---
>
> ### **Cons5: Suggestion for the discussion on real-world usability and robustness.**
>
> **R5:** Thank you for your suggestion regarding practical constraints. We appreciate the opportunity to clarify that **runtime and data usage considerations** are already discussed in Section 4.2, specifically in Tables 3–6 for each attack. Taking ObjectRep as an example, we observe that the editing-based method EvilEdit is the most efficient, requiring only 16.59 seconds and no additional data to achieve an ASR of 61.10%. In contrast, the training-based method TPA (RickRolling) achieves a higher ASR (96.80%, which is 35.70% higher than EvilEdit), but at the cost of 286.89 seconds and 25,600 training samples. This illustrates a clear trade-off between attack effectiveness and resource consumption.
>
> Regarding **robustness**, since adversarial adaptations and adaptive attacks are typically designed on a case-by-case basis for each defense, they are supposed to be discussed in the original papers that propose those defenses. Here, we add a simple word-level perturbation to the input text to preliminarily assess the robustness of the defense. Specifically, we randomly replace 5%, 10%, and 50% of the words with their synonyms and evaluate the T2I backdoor detection method, T2IShield, on the perturbed input. The results are presented below (Tab. 2). The decreasing performance from the increasing perturbations indicates that the defense is fragile to textual perturbations, where simple synonym replacement can make a big difference. Due to time constraints, we will include a more thorough discussion in the revised version of the paper.
>
> **Tab. 2. Defense comparison of T2IShield with regular textual perturbation. (%)**
>
> |T2IShield|Original|Original|Original|5% Perturbed|5% Perturbed|5% Perturbed|10% Perturbed|10% Perturbed|10% Perturbed|50% Perturbed|50% Perturbed|50% Perturbed|
> -|-|-|-|-|-|-|-|-|-|-|-|-
> Attacks|Precision|Recall|F1-Score|Precision|Recall|F1-Score|Precision|Recall|F1-Score|Precision|Recall|F1-Score
> Object-Backdoor (BadT2I)|**26.25**|**12.60**|**17.03**|19.75|9.40|12.74|13.30|6.20|8.46|2.89|1.40|1.89
> EvilEdit|**40.00**|**27.20**|**32.38**|35.21|23.80|28.40|27.79|18.40|22.14|4.61|3.20|3.78
>
> ---
>
> [1] How to Backdoor Diffusion Models?, CVPR 2023.
>
> [2] Rickrolling the Artist: Injecting Backdoors into Text Encoders for Text-to-Image Synthesis, ICCV 2023.
>
> [3] TIFA: Accurate and Interpretable Text-to-Image Faithfulness Evaluation with Question Answering, ICCV 2023.
>
> [4] T2I-CompBench: A Comprehensive Benchmark for Open-world Compositional Text-to-image Generation, NeurIPS 2023.

---

> > ### Author Response · Authors · 2025-08-05
> >
> > Dear Reviewer Yt6A:
> >
> > We would like to express our sincere gratitude for your valuable insights and suggestions on our work.
> >
> > We have tried our best to address the concerns and queries you raised during the rebuttal process. We would greatly appreciate knowing whether our responses have effectively resolved your concerns. Your feedback will be essential in further improving the quality of our work.
> >
> > Sincerely,
> >
> > Authors

---

> > ### Comment · Reviewer_Yt6A · 2025-08-05
> >
> > Thanks for the response. Most of my concerns are addressed. I would increase my rating to borderline accept.

---

> > > ### Author Response · Authors · 2025-08-05
> > >
> > > Dear Reviewer Yt6A,
> > >
> > > We sincerely appreciate your thoughtful response and the time you've dedicated to reviewing our paper. We will incorporate your suggestions and insights into the revised manuscript. Thank you once again for your thorough review and positive evaluation.
> > >
> > > Sincerely,
> > >
> > > Authors

---

### Official Review · Reviewer_wD1L · 2025-07-03

**Rating:** 5
**Confidence:** 5

**Summary:**

This work provides a comprehensive benchmark of the backdoor attack for diffusion models, which involves 9 SOTA attack methods and 4 SOTA defense methods. Beyond the methods, they also unified the evaluation metrics scattered across previous works. Their evaluation results also study the Stable Diffusion (SD) v2.0, which has never been checked by previous works. The main manuscript reveals the status of the existing attack methods under different scenarios, and abundant results are shown in the Appendix.

**Additional Feedback:**

While I can reproduce the similar results reported in this paper, I am concerned about the reproducibility of T2IShield in BackdoorDM. I am willing to increase my scores if the authors could provide supporting reproducibility results of T2IShield.

**Dataset Code Accessibility:**

Partly

**Dataset Code Comments:**

1. Please add `pip install peft` in your *requirement.txt* file to prevent the error from running VillanDiffusion. (`ValueError: PEFT backend is required for saving LoRAs using the `save_attn_procs()` method.`)
2. Please provide the step-by-step script or tutorial to reproduce Table 1 in the paper of T2IShield with `mignneko` trigger. As I mentioned in **Limitations Weaknesses point 1**, I cannot reproduce the Precision/Recall/F1-Score.


Here are my values and original values reported in T2IShield.
- Precision: 50.03% vs 61.0%
- Recall: 75.20% vs 100.0%
- F1-score: 60.09% vs 75.8%

The following are my steps to generate results:
1. `time python ./attack/t2i_gen/villan_diffusion_cond/villan_cond.py --bd_config "./attack/t2i_gen/configs/bd_config_fix.yaml" --pretrained_model_name_or_path "CompVis/stable-diffusion-v1-4"`
2. `time python ./defense/model_level/t2ishield/t2ishield.py --base_config "./defense/model_level/configs/t2ishield.yaml" --backdoor_method "villandiffusion_cond" --backdoored_model_path "./results/villandiffusion_cond_v1-5/villandiffusion_cond_trigger-mignneko_target-cat/" --bd_config "./attack/t2i_gen/configs/bd_config_fix.yaml"`
3. I check the results in `./results/villandiffusion_cond_v1-5/villandiffusion_cond_trigger-mignneko_target-cat/detected_backdoor_samples.txt` to calculate above values.

3. Please update README.md about **Run the defense.** example.
Original version.
```shell
# Defense example
python ./attack/t2i_gen/eviledit/eviledit.py \
    --base_config 'attack/t2i_gen/configs/base_config.yaml' \
    --bd_config 'attack/t2i_gen/configs/bd_config_objectRep.yaml' \
    --model_ver 'sd15' \
    --device 'cuda:0'
```
I think that it should be this command.
```shell
python ./defense/model_level/t2ishield/t2ishield.py --base_config "./defense/model_level/configs/t2ishield.yaml" --backdoor_method "eviledit" --backdoored_model_path "./results/eviledit_sd15/eviledit_trigger-beautifuldog_target-cat.pt" --bd_config "./attack/t2i_gen/configs/bd_config_objectRep.yaml"
```

**Ethical Considerations:**

No, there are no or only very minor ethics concerns

**Final Justification:**

While several reviewers proposed issues regarding the novelty and depth of the insights, I consider that this benchmark would help us to have a more *fair environment* to compare existing SOTA attacks and defense methods in text-to-image.
Therefore, I would like to increase my score to continually encourage the author to grow a solid benchmark.

**Limitations Weaknesses:**

1. This work does not include detection evaluation, such as precision/recall/F1-score, for model-level defense. The critical part of assessing T2IShield is to evaluate its detection capability. I strongly suggest that the authors give more experimental results or tools. If no, please provide a sensible reason to rebut me.
2. Some defense methods, such as UFID and PureDiffusion, are mentioned but not implemented. This is just my wish list that hopes the authors will implement it soon.

**Strengths Contributions:**

This work contributes to the complete ecosystem--BackdoorDM Python package to allow practitioners to easily investigate the backdoor methods for diffusion models. I trust the good impact of this package on the research community.

The paper's presentation is clear with detailed settings and well-organized categories of existing attack and defense methods. The critical issue of this field is the evaluation of different methods under different scenarios, and the authors provide a systematic research framework connected with related works from the literature. In particular, introducing a multimodal large language model to conduct the evaluation is a good example to utilize the emerging tools to advance the field.

---

> ### Author Rebuttal · Authors · 2025-07-31
>
> Thank you very much for your positive appraisal and great interest in our paper. We are encouraged by your positive comments on our **practical value**, **good impact**, **clear presentation**, **well-organized categories**, and **advanced MLLM evaluation**. We hope the following responses could help alleviate your concerns.
>
> ---
>
> ### **Cons1: Suggestion for assessing the detection capability of T2IShield defense.**
>
> **R1:** Thanks for your valuable suggestion. We are pleased to share our reproduction results on precision/recall/F1-score with you. Below is a step-by-step guidance for running VillanDiffusion and subsequently applying T2IShield detection on SDv1.4:
>
> 1. **Run VillanDiffusion with `mignneko` trigger and Stable Diffusion v1-4.**
>
>     Thanks for mentioning the ‘PEFT backend’ issue in the code comment. We have solved this issue by specifying a new version of diffusers in the latest ‘install.sh’ file. After setting up the environment, you can refer to the following steps to run VillanDiffusion:
>
>     - Step 1: Edit `attack/t2i_gen/configs/bd_config_fix.yaml`, set `caption_trigger` to `TRIGGER_MIGNNEKO`. In our experiment, we set the trigger to `TRIGGER_LATTE_COFFEE` by default, following the literature. You can also choose other triggers listed in `attack/t2i_gen/villan_diffusion_cond/caption_dataset.py`.
>     - Step 2: Run the following command to run VillanDiffusion attack:
>
>         ```
>         python attack/t2i_gen/villan_diffusion_cond/villan_cond.py --pretrained_model_name_or_path CompVis/stable-diffusion-v1-4 --bd_config ./attack/t2i_gen/configs/bd_config_fix.yaml
>         ```
>
>         This will generate a result directory `results/vilandiffusion_cond_sd14` , which contains the attacked checkpoint. After the attack is done, you can test T2IShield on it.
>
> 2. **Run T2IShield to detect backdoor samples (reproduce Precision/Recall/F1-Score).**
>
>     Our reproduced version of T2IShield is divided into three substeps. Running substep 1 is sufficient to detect benign and backdoor samples.
>
>     - Step 1: Add the calculation of the three metrics (Precision, Recall and F1-Score) after the code of substep1 in `defense/model_level/t2ishield/t2ishield.py`(i.e., after getting benign_samples and backdoor_samples). The calculation code is listed followed:
>
>         ```python
>         # Calculate precision, recall, and F1 score
>         true_positives = sum(1 for sample in backdoor_samples if sample in bd_prompts)
>         false_positives = len(backdoor_samples) - true_positives
>         false_negatives = len(bd_prompts) - true_positives
>
>         precision = true_positives / len(backdoor_samples) if len(backdoor_samples) > 0 else 0
>         recall = true_positives / len(bd_prompts) if len(bd_prompts) > 0 else 0
>         f1_score = 2 * precision * recall / (precision + recall) if (precision + recall) > 0 else 0
>         ```
>
>     - Step 2: Edit `defense/model_level/configs/t2ishield.yaml`, change the following settings:
>
>         ```
>         model_ver: 'sd14'
>         train_dataset: 'celeba_hq'
>         caption_column: 'text'
>         image_column: 'image'
>         ```
>
>     - Step 3: Run the following command to test T2IShield defense:
>
>         ```
>         python defense/model_level/t2ishield/t2ishield.py --backdoor_method villandiffusion_cond --backdoored_model_path results/villandiffusion_cond_sd14 --execute_steps 1
>         ```
>
>         This will generate a defense directory in the result directory, which is `results/vilandiffusion_cond_stable-diffusion-v1-4/defense/t2ishield`. You can check `detected_backdoor_samples.txt` or logs for detection results, which will also be output in the terminal.
>
> 3. **Our reproduced results.**
>
>     By executing the above steps, we receive similar results to those in the original paper of T2IShield (i.e., Precision: 61%, Recall: 100%, F1-Score: 75.8%) [1]. Our reproduced results are listed as follows:
>
>     **Tab. 1. Detection performance of T2IShield reproduced in BackdooDM.**
>
>     | **Trigger** | **Precision (%)** | **Recall (%)** | **F1-Score (%)** |
>     | --- | --- | --- | --- |
>     | `TRIGGER_LATTE_COFFEE` | 56.88 | 100 | 72.52 |
>     | `TRIGGER_MIGNNEKO` | 60.46 | 100 | 75.36 |
>
> We hope the above guidance could help your research and alleviate your concerns.
>
> ---
>
> ### **Cons2: Wish list for implementing UFID and PureDiffusion.**
>
> **R2:** Thank you for mentioning the specific defense methods you need. They are actually in our TODO list also. However, the reason why we have not yet implemented them is that we have some problems in reproducing these methods. Specifically, although UFID is an open-source project, its official GitHub repository is not complete. As a simple example, by following their instruction to run "bash run_conditional.sh" for their demo example with the Pokemon-caption dataset, we found that it just conducts backdoor injection, sampling, and basic quality assessment, without multiple perturbations for each input and the defensive detection as suggested in the paper. We have contacted the original author and confirmed this problem with them, and we are currently working closely to address this issue. For PureDiffusion, as it is closed-source, we have tried to ask its authors whether they could share the source code with us to facilitate an efficient and faithful reproduction, but have, unfortunately, not received any responses yet. If this case continues, we will try to reproduce the code ourselves.
>
> Except for the above-mentioned defenses, we also notice a newly released backdoor detection method, DAA [2], in a preprint paper after the submission date of NeurIPS’25. For your reference, we reproduce it in BackdoorDM as the new SOTA detection method. The results are shown below (Tab. 2). Despite the improvement of DAA compared to T2IShield, we can also observe its limitation on other Text-to-Image (T2I) attacks except for VillanDiffusion. The reproduced code will be released in the subsequent code version.
>
> **Tab. 2. Detection performance comparison of DAA on three T2I attacks. (%)**
>
> |  | **VillanDiffusion** | **VillanDiffusion** | **VillanDiffusion** | **Object-Backdoor (BadT2I)** | **Object-Backdoor (BadT2I)** | **Object-Backdoor (BadT2I)** | **EvilEdit** | **EvilEdit** | **EvilEdit** |
> | --- | --- | --- | --- | --- | --- | --- | --- | --- | --- |
> |  | Precision | Recall | F1-Score | Precision | Recall | F1-Score | Precision | Recall | F1-Score |
> | T2IShield  | 56.88 | 100 | 72.52 | 26.25 | 12.60 | 17.03 | 40.00 | 27.20 | 32.38 |
> | DAA-I | 98.04 | **90.00** | **93.85** | **51.27** | **20.20** | **28.98** | **60.31** | **31.60** | **41.47** |
> | DAA-S | **98.83** | 33.80 | 50.37 | 50.79 | 6.40 | 11.37 | 52.24 | 7.00 | 12.35 |
>
> ---
>
> [1] T2IShield: Defending Against Backdoors on Text-to-Image Diffusion Models, ECCV 2024.
>
> [2] Dynamic Attention Analysis for Backdoor Detection in Text-to-Image Diffusion Models, Arxiv 2025.

---

> > ### Author Response · Authors · 2025-08-05
> >
> > Dear Reviewer wD1L:
> >
> > We would like to express our sincere gratitude for your great interest and valuable suggestions on our work.
> >
> > During the rebuttal process, we have implemented detection evaluation, added a SOTA defense baseline, and provided a step-by-step script to reproduce T2IShield with the mignneko trigger. The relevant code has been updated in our codebase. Due to NeurIPS policy, we are unable to include external links here.
> >
> > You kindly mentioned that you are willing to increase your score if supporting reproducibility results of T2IShield are provided. We would greatly appreciate knowing whether our updated implementation and additional results have effectively resolved your reproducibility concerns.
> >
> > Your feedback will be essential in further improving the quality of our work.
> >
> > Sincerely,
> >
> > Authors

---

> > > ### Comment · Reviewer_wD1L · 2025-08-05
> > >
> > > Thanks for your feedback and step-by-step instructions.
> > >
> > > While several reviewers proposed issues regarding the novelty and depth of the insights, I would like to thank the author for their effort in the backdoor attack field, which helps us to have a more *fair environment* to compare existing SOTA attacks and defense methods in text-to-image.
> > > Especially, after the author gives more information on MLLM evaluation to allay some reviewers' concerns, I have more confidence in using it in future work.
> > >
> > > Therefore, I would like to increase my score to continually encourage the author to grow a solid benchmark. Thanks.

---

> > > > ### Author Response · Authors · 2025-08-05
> > > >
> > > > Dear Reviewer wD1L,
> > > >
> > > > We sincerely appreciate your encouragement and the time you've dedicated to reviewing our paper. It is particularly rewarding to know that our benchmark is useful to researchers like yourself, and we are committed to continuously improving its quality and utility for the research community in the future. Thank you once again for your thorough review and positive evaluation.
> > > >
> > > > Sincerely,
> > > >
> > > > Authors

---

### Official Review · Reviewer_NH5s · 2025-07-07

**Rating:** 4
**Confidence:** 4

**Summary:**

This paper presents backdoor benchmarking for diffusion models, called BackdoorDM. It focuses on characterizing different backdoor attack and defense methods for diffusion models and introduces a unified evaluation approach using a multimodal large language model (MLLM).

Specifically, this work proposes a unified framework for modeling various backdoor attacks and target types in diffusion models by setting appropriate hyperparameter values. It also presents a novel evaluation method that leverages an MLLM to assess backdoor targets and provide image-level evaluations.

**Dataset Code Accessibility:**

Yes

**Ethical Considerations:**

No, there are no or only very minor ethics concerns

**Final Justification:**

I have upgraded the score from 3 to 4.

The author(s) addressed the issues I raised and promised to effect relevant changes in the final version.

**Limitations Weaknesses:**

Presentation: Figure 2 is very difficult to read and interpret. Since it is key to understanding the unified framework proposed in the paper, its current state significantly hampers clarity. Additionally, the paper introduces new terminology related to types of backdoor targets. Including an illustrative figure to depict these target types would greatly improve comprehension.

Language and Formatting: There are multiple instances where uppercase letters appear incorrectly at the beginning of words or phrases (e.g., Lines 168, 171, 189, 190). These should be revised for consistency and proper sentence structure.

Supporting Evidence: Most of the core results are currently presented in the appendices. To strengthen the paper's main claims, consider distilling and integrating key findings into the main body of the paper.

**Strengths Contributions:**

Significance: The potential impact of this work is substantial, given the widespread popularity of diffusion models. The benchmark introduced in this study provides valuable insights that will help researchers better understand existing backdoor attack and defense strategies within this domain.

Novelty: The unified metric leveraging multimodal large language models (MLLMs) highlights the innovative contribution of this work. It offers a powerful tool for advancing research on backdoor attacks and defenses tailored to diffusion models.

---

> ### Author Rebuttal · Authors · 2025-07-31
>
> Thank you very much for your thoughtful review and positive comments on our work. We are greatly encouraged by your recognition of our **substantial potential impact**, **valuable insights**, and **innovative MLLM evaluation**. We hope the following responses could help clarify the potential misunderstandings and alleviate your concerns.
>
> ---
>
> ### **Cons1: Suggestion for the interpretation of Figure 2 and additional illustrative figures for backdoor targets.**
>
> **R1:** Thanks for your valuable suggestions. Figure 2 exhibits the step-by-step process of how Equation (1) is constructed for better understanding, where each formula letter is clearly stated at the bottom. Starting from the Gaussian noise and text input on two sides, the goal of backdoor injection is to align the three processed targets from the trainable backdoored DM ($\hat{M}$) and the frozen benign DM ($M$) in the middle. We introduce the details as follows:
>
> > Firstly, the noise and text input are modified according to the desired trigger–target pair. For example, by adding the word *‘trigger’* to $\hat{M}$’s text input and ‘black and white photo’ to $M$’s text input, we can align the two inputs to facilitate the injection of a backdoor into $\hat{M}$. Secondly, we use $\hat{M}$ and $M$ to process the transformed inputs and extract the corresponding target hidden states. Then, we can also introduce target image sources $S$ on the frozen side (e.g., the right-side model) to further specify the desired target. Finally, the hidden states from both sides are aligned to inject the malicious mapping into the left-side model, which is trainable and backdoored.
> >
>
> We will add a detailed interpretation at line 122, following the reference to Figure 2. For the illustrative figures, we will include 1–3 examples for each backdoor target in the revised version.
>
> ---
>
> ### **Cons2: Reminder of the typos.**
>
> **R2:** Thank you very much for your careful review. We will carefully proofread our paper and revise all typos in the next version. For your mentioned cases, we will correct them as follows:
>
> > Line 168: Input-level → Input level;
> >
> > Line 171: Model-level → Model level;
> >
> > Line 189: Run Time → Run time;
> >
> > Line 190: Data Usage → Data usage.
>
> ---
>
> ### **Cons3: Suggestions for integrating the key findings into the main text.**
>
> **R3:** Thanks for your valuable suggestion on the presentation of the key findings. We sincerely apologize for postponing some important findings to the appendix due to the space limit of the main text. In the revised version, we will rearrange the space and summarize some key findings in the Introduction Section. Here, we summarize the findings from the perspective of attacks, defenses, and visualizations as follows for your reference:
>
> > Our extensive experiments yield several important findings, encompassing the results of current attacks, defenses, and the visualization of backdoored models.
> > - **Attacks**: 1) Among the existing backdoor targets, ImageFix and StyleAdd achieve higher ASRs, indicating their greater risk in practical applications. 2) The ObjectRep target attracts the most research attention; however, current attack methods fail to perform consistently across various evaluation metrics. 3) Non-conflicting backdoor targets may be more vulnerable to attack.
> > - **Defenses**: The current performance of model-level defenses is limited for both unconditional DMs (Elijah) and text-to-image (T2I) DMs (T2IShield and Textual Perturbation), whereas the input-level method TERD demonstrates greater effectiveness.
> > - **Visualizations**: 1) *Assimilation Phenomenon*: The ImageFix target exhibits a more pronounced assimilation pattern, while other target types may not show such behavior, indicating the limitations of using attention maps for backdoor visualization.  2) *Activation Norm*: Certain neurons in backdoored DMs consistently show significant differences in activation norms between clean and poisoned samples, suggesting a potential defense strategy for identifying backdoored neurons. 3) *Pre-Activation Distribution*: A clear bimodal distribution is observed in certain neurons, highlighting the need for further investigation into neuron functionality for backdoor defense.

---

> > ### Author Response · Authors · 2025-08-05
> >
> > Dear Reviewer NH5s:
> >
> > We would like to express our sincere gratitude for your valuable insights and suggestions on our work.
> >
> > We have tried our best to address the concerns and queries you raised during the rebuttal process. We would greatly appreciate knowing whether our responses have effectively resolved your concerns. Your feedback will be essential in further improving the quality of our work.
> >
> > Sincerely,
> >
> > Authors

---

> > ### Author Response · Authors · 2025-08-08
> >
> > Dear Reviewer NH5s:
> >
> > Thanks again for your valuable suggestions. As the end of the discussion period is approaching, we would like to kindly ask again for your concerns about our paper. Are there still any unsolved doubts for you?
> >
> > Your help is greatly appreciated. We are eagerly waiting for your feedback before the end.
> >
> > Sincerely,
> >
> > Authors

---

### Official Review · Reviewer_aDMm · 2025-07-07

**Rating:** 5
**Confidence:** 5

**Summary:**

This paper proposes BackdoorDM, the first comprehensive benchmark for evaluating backdoor attacks and defenses in Diffusion Models (DMs). Unlike prior studies focused on discriminative models, this benchmark is tailored for the unique challenges of generative DMs, particularly for unconditional generation and text-to-image (T2I) tasks. BackdoorDM consists of: 9 state-of-the-art (SOTA) attack methods (e.g., BadDiffusion, EvilEdit, BadT2I). 4 SOTA defense strategies (e.g., Elijah, T2IShield). 3 visualization tools (e.g., activation norm differences, pre-activation distributions)

**Dataset Code Accessibility:**

Yes

**Ethical Considerations:**

No, there are no or only very minor ethics concerns

**Final Justification:**

After considering the authors’ response and the comments from other reviewers, I have raised my score to 5.

**Limitations Weaknesses:**

While the use of MLLM-based evaluation introduces a novel perspective, it may incur significant computational overhead and suffer from inconsistency due to its dependence on natural language understanding and the variability inherent in prompt design.

The study evaluates only four defense methods, omitting several state-of-the-art and emerging approaches. This limited scope may restrict the generalizability and comprehensiveness of the findings.

The paper lacks clear insights or novel takeaways for the research community. Many of the presented results and discussions offer limited depth or originality, making it difficult to identify substantial contributions or new understanding derived from the work.

**Strengths Contributions:**

(1) First systematic benchmark specifically for diffusion backdoors, a rapidly emerging threat.
(2) Provides a clear and extensible formulation (Equation 1) for backdoor learning across diffusion paradigms. Offers a taxonomy that clarifies distinctions in trigger mechanisms, targets, and modalities.
(3) Proposes a novel evaluation method using LLMs (e.g., GPT-4o) to assess ASR, PSR, and ACC, improving upon the limitations of prior classifier-based metrics.

---

> ### Author Rebuttal · Authors · 2025-07-31
>
> We sincerely thank you for your valuable feedback and for recognizing the significance of our work. We are encouraged by your positive comments on our **first systematic benchmark**, **clear and extensible formulation**, and **novel evaluation method**. We hope the following responses could help alleviate your concerns.
>
> ---
>
> ### **Cons1: Concern about the computational overhead and consistency of MLLM evaluation.**
>
> **R1:** Thanks for your concern. We acknowledge that MLLM evaluation incurs high **computational costs**; however, this is a common challenge across all large-model-based evaluation methods [1,2]. To meet this challenge, on one hand, we have implemented the traditional evaluation methods, as illustrated in Section 3.4, to ensure that our benchmark can be used under the lightweight-overhead setting; on the other hand, we also assess the MLLM evaluation with backbones in different sizes, as discussed in Appendix C.1, to offer more options with size-performance trade-off for the users. Finally, this issue will be alleviated by increasingly mature LLM light-weighting techniques and progressively lower computational costs.
>
> Similarly, the potential **consistency issue** is also considered to be a common challenge in relevant research [1,2]. To ensure the reproducibility of our evaluation results, we offer the MLLM prompts in Appendix B.7, MLLM versions in Appendix C.1, and experimental settings in Appendix B.6. Here, we further test the performance consistency of MLLM evaluations (based on GPT4-o and Qwen2-VL-72b) across three random seeds, i.e., 0, 100, 678. The results are illustrated below (Tab. 1). We can observe that most of the results range within 1%, indicating the high consistency of using either closed-source (GPT-4o) or open-source (Qwen2.5-VL) MLLMs.
>
> **Tab. 1. MLLM evaluation results comparison on different random seeds. (%)**
>
> | ObjectRep | MLLM | GPT-4o | GPT-4o | GPT-4o | GPT-4o | Qwen2.5-VL-72b | Qwen2.5-VL-72b | Qwen2.5-VL-72b | Qwen2.5-VL-72b |
> | --- | --- | --- | --- | --- | --- | --- | --- | --- | --- |
> | Attacks | Random Seed | 0 | 100 | 678 | **Range** | 0 | 100 | 678 | **Range** |
> | TPA (RickRolling) | ACC | 82.61 | 83.42 | 83.41 | **0.81** | 78.29 | 78.17 | 78.33 | **0.16** |
> | TPA (RickRolling) | ASR | 96.80 | 96.80 | 96.80 | **0.00** | 96.90 | 96.90 | 96.90 | **0.00** |
> | TPA (RickRolling) | PSR | 5.01 | 4.43 | 5.50 | **1.07** | 2.86 | 3.16 | 3.22 | **0.36** |
> | DB (Paas) | ACC | 70.89 | 71.29 | 70.87 | **0.42** | 64.70 | 64.51 | 64.70 | **0.19** |
> | DB (Paas) | ASR | 51.40 | 51.20 | 51.30 | **0.20** | 51.45 | 51.50 | 51.70 | **0.25** |
> | DB (Paas) | PSR | 60.45 | 59.64 | 60.22 | **0.81** | 52.06 | 52.29 | 52.02 | **0.27** |
> | EvilEdit | ACC | 83.82 | 83.83 | 83.01 | **0.82** | 79.62 | 79.85 | 79.46 | **0.39** |
> | EvilEdit | ASR | 61.00 | 60.60 | 61.10 | **0.50** | 59.50 | 60.20 | 59.70 | **0.70** |
> | EvilEdit | PSR | 84.30 | 83.89 | 85.25 | **1.36** | 75.94 | 76.52 | 75.55 | **0.97** |
>
> ---
>
> ### **Cons2: Concern about the comprehensiveness of findings on defense methods.**
>
> **R2:** Thanks for your concern about the defense methods. We noticed that several defense methods were released in preprint versions [3-6] near the date of NeurIPS’25 submission, while most of their code **had not yet been made publicly available**. Due to the time limit, we find it hard to reproduce them from scratch. Here, we add the newest backdoor detection method DAA [6], including both DAA-I and DAA-S versions, to illustrate the consistency of our findings. The performance comparison is shown below (Tab. 2), indicating that although DAA-I performs comparably in VillanDiffusion, most of the results fail on one or more detection metrics among all attacks. These results also exhibit the limitation of the current SOTA DAA defense, as we have concluded in Appendix C.4. We will add more discussion on the new defenses in the revised version.
>
> **Tab. 2. Detection performance comparison of DAA on three T2I attacks. (%)**
>
> |  | **VillanDiffusion** | **VillanDiffusion** | **VillanDiffusion** | **Object-Backdoor (BadT2I)** | **Object-Backdoor (BadT2I)** | **Object-Backdoor (BadT2I)** | **EvilEdit** | **EvilEdit** | **EvilEdit** |
> | --- | --- | --- | --- | --- | --- | --- | --- | --- | --- |
> |  | Precision | Recall | F1-Score | Precision | Recall | F1-Score | Precision | Recall | F1-Score |
> | T2IShield | 56.88 | 100 | 72.52 | 26.25 | 12.60 | 17.03 | 40.00 | 27.20 | 32.38 |
> | DAA-I | 98.04 | **90.00** | **93.85** | **51.27** | **20.20** | **28.98** | **60.31** | **31.60** | **41.47** |
> | DAA-S | **98.83** | 33.80 | 50.37 | 50.79 | 6.40 | 11.37 | 52.24 | 7.00 | 12.35 |
>
> ---
>
> ### **Cons3: Concern about the insights and contribution of our work.**
>
> **R3:** Thanks for your insightful concern about the insights and contribution of our work. We would like to kindly point out that we have highlighted several insights and takeaways in boldface at the end of each experiment, most of which focus on bringing new understanding to the current and future research directions. For a better overview, we will summarize our insights and takeaways in the Introduction Section of the revised version. Here is a brief version:
>
> > Our extensive experiments yield several important findings, encompassing the results of current attacks, defenses, and the visualization of backdoored models.
> > - **Attacks**: 1) Among the existing backdoor targets, ImageFix and StyleAdd achieve higher ASRs, indicating their greater risk in practical applications. 2) The ObjectRep target attracts the most research attention; however, current attack methods fail to perform consistently across various evaluation metrics. 3) Non-conflicting backdoor targets may be more vulnerable to attack.
> > - **Defenses**: The current performance of model-level defenses is limited for both unconditional DMs (Elijah) and text-to-image (T2I) DMs (T2IShield and Textual Perturbation), whereas the input-level method TERD demonstrates greater effectiveness.
> > - **Visualizations**: 1) *Assimilation Phenomenon*: The ImageFix target exhibits a more pronounced assimilation pattern, while other target types may not show such behavior, indicating the limitations of using attention maps for backdoor visualization.  2) *Activation Norm*: Certain neurons in backdoored DMs consistently show significant differences in activation norms between clean and poisoned samples, suggesting a potential defense strategy for identifying backdoored neurons. 3) *Pre-Activation Distribution*: A clear bimodal distribution is observed in certain neurons, highlighting the need for further investigation into neuron functionality for backdoor defense.
> >
>
> Except for the summarized experimental insights above, our contribution as the first benchmark in diffusion backdoor is also recognized by the reviewers from the research community, including 1) *the unified formulation that may inspire future attack or defense designs*. (from reviewer mCTK); 2) *the benchmark will help researchers better understand existing backdoor attack and defense strategies within this domain. The potential impact is substantial* (from reviewer NH5s); 3) *the good impact of this package on the research community is worth trusting.* (from reviewer wD1L).
>
> We sincerely hope that our benchmark could contribute to the research community and help build a trustworthy AIGC community.
>
> ---
>
> [1] TIFA: Accurate and Interpretable Text-to-Image Faithfulness Evaluation with Question Answering, ICCV 2023.
>
> [2] T2I-CompBench: A Comprehensive Benchmark for Open-world Compositional Text-to-image Generation, NeurIPS 2023.
>
> [3] A Dual-Purpose Framework for Backdoor Defense and Backdoor Amplification in Diffusion Models, Arxiv 2025.
>
> [4] NaviDet: Efficient Input-level Backdoor Detection on Text-to-Image Synthesis via  Neuron Activation Variation, Arxiv 2025.
>
> [5] Backdoor Defense in Diffusion Models via Spatial Attention Unlearning, Arxiv 2025.
>
> [6] Dynamic Attention Analysis for Backdoor Detection in Text-to-Image Diffusion Models, Arxiv 2025.

---

> > ### Comment · Reviewer_aDMm · 2025-08-01
> > **Response**
> >
> > Dear authors,
> >
> > Thank you for your detailed response. I appreciate the efforts you made to address the reviewers’ concerns, including mine.
> >
> > After carefully considering the feedback from all reviewers and your clarifications, I acknowledge that some of my earlier concerns have been addressed. However, similar to Reviewer mCTK, I still have some reservations regarding the novelty and depth of the insights presented in the paper. Specifically, the discussions around Activation Norm and Pre-Activation Distribution align with existing understandings in the backdoor literature and do not appear to provide substantial new insights.
> >
> > Given this, I will maintain my original evaluation of Borderline Accept.
> >
> > Best regards,

---

> > > ### Author Response · Authors · 2025-08-01
> > >
> > > Dear Reviewer aDMm,
> > >
> > > Thank you for your timely response and for specifying your remaining concerns.
> > >
> > > We acknowledge your concern regarding the ‘similar’ observations from Activation Norm and Pre-Activation Distribution between the traditional backdoor literatures and ours. However, we are in different settings, e.g., the previous observations are based on discriminative models, while we focus on diffusion models (DMs). It would be inappropriate to overlook the huge gap between the inference paradigms (as illustrated in Figure 1) and adopt the previous conclusions directly. Therefore, our contribution involves filling this gap and providing more DM-specific insights.
> > >
> > > For example, regarding Activation Norm, we illustrate and highlight the DM-specific temporal dynamics in Figure 5 (Appendix C.7). It can be observed that, with increasing inference steps, some neurons exhibiting significantly larger norm differences at the beginning gradually decrease over time in the unconditional DM, while the completely opposite phenomenon is observed in T2I DM, i.e., neurons exhibit larger norm differences as time progresses. Similarly, this temporal characteristic also brings a DM-specific perspective when observing DMs via Pre-Activation Distribution.
> > >
> > > Building on these new insights, we encourage more further research to deepen the understanding of this emerging diffusion backdoor threat.
> > >
> > > Sincerely,
> > >
> > > Authors

---

> > > > ### Comment · Reviewer_aDMm · 2025-08-05
> > > > **Response**
> > > >
> > > > After considering the authors’ response and the comments from other reviewers, I have raised my score to 5.

---

> > > > > ### Author Response · Authors · 2025-08-06
> > > > >
> > > > > Dear Reviewer aDMm,
> > > > >
> > > > > We sincerely appreciate your thoughtful response and the time you've dedicated to reviewing our paper. Your feedback is incredibly valuable, and we will carefully incorporate your suggestions—particularly the DM-specific insights—into the revised manuscript to strengthen our contribution. It is truly encouraging to receive such constructive and supportive input, and we are grateful for your positive evaluation. Thank you once again for your time and expertise
> > > > >
> > > > > Sincerely,
> > > > >
> > > > > Authors

---

### Official Review · Reviewer_mCTK · 2025-07-20

**Rating:** 4
**Confidence:** 2

**Summary:**

This paper points out that backdoor attacks in diffusion models are much more diverse than indiscriminative models — not only in how the triggers are injected, but also in what the targetsactually look like. Such diversity makes fair comparisons harder. To To address this, the paperproposes to use an MLLM to evaluate a range of existing attacks across various kinds ofmetrics, instead of training lots of separate binary classifiers or utilizing unsuitable adapationsas existing fragmented evaluations.

**Dataset Code Accessibility:**

Yes

**Ethical Considerations:**

No, there are no or only very minor ethics concerns

**Final Justification:**

I have read the response which solves my concerns. Therefore, I decide to raise my score.

**Limitations Weaknesses:**

1. I think the writing could be better, especially the organiization of experiments. It would be better to summarize some key points from the appendix into the main text, instead of simply listing. It feels a bit scattered from currentmanuscript.
2. For experiment analysis, it might be desired to see (or highlight) whether this benchmark leads to any new or counterintuitive conclusionis about existing attacks-something that previous evaluations might have missed.
3. My biggest concern is about whether using an MLLM alcone is really sufficient-or superior-to replace specifically trained classifiers. Yes, MLLMs are powerful and commonly used in T21 tasks and much more convenient than training multiple classifiers, but it's still unclear whether they're actually reliable enough at detecting backdoor behaviors, especially for subtle or pixel-level triggers. Therefore, a more
detailed (which should be highlighted) comparison of its albility with previous evaluation methods might be desired.

For instance, small patches or imperceptible triggers mightt fool the diffusion model but get missed by the MLLM. So it'd be great to at least compare GPT-4o's evaluation with a simple binary classifier trained to detect triggered vs cleaan outputs. That would make the trust in MLLM-based evaluation much more solid.If GPT-40 misses attacks the classifier catches (or vice versa), that would be importantto know, and it would help clarify the strengths and limits of using MLLMs for this task

**Strengths Contributions:**

1. This paper is well-motivated and addresses the troublesome, unfairness and fragmented comparisons of current backdoor attacks evaluations of for DMS.
2. It organizes and formalizes existing attack methods within a clean and unified framework pretty well, which helps clarify the current development and might inspire future attack or defense designs.
3. The experiments are comprehensive and quite thorough, with various attack and defense methods, as well as comprehensive metrics aand visualizations

---

> ### Author Rebuttal · Authors · 2025-07-31
>
> Thank you very much for your careful review of our paper and constructive comments. We are encouraged by your positive comments on our **well-motivated study**, **fair evaluation**, **clean and unified framework**, and **comprehensive experiments**. We hope the following responses could help clarify the potential misunderstandings and alleviate your concerns.
>
> ---
>
> ### **Cons1: Suggestion for summarizing the experiment key points in the main text.**
>
> **R1:** We sincerely appreciate your constructive suggestion for a better presentation. We would like to kindly point out that several appendix experiments are introduced following the corresponding main-text contents for smooth reading, e.g., C.4 in Section 3.3, C.1 in Section 3.4, and C.7 in Section 3.6. In the revised version, we will summarize the key points to refine Section 4.3 of the main text. Here is a brief version:
>
> > For more experiments in Appendix C, we summarize their key points below:
> >
> > (C.1) Further analysis of MLLM evaluation. We comprehensively assess the proposed MLLM evaluation on 4 more open-source MLLMs and human-labeled data.
> >
> > (C.2) More results for different datasets. We investigate the impact of using high-resolution images for attacks using CelebA-HQ.
> >
> > (C.3) Effect of poisoning ratio. We test the effect of different poisoning ratios from 0.1 to 0.9.
> >
> > (C.4) Defense results. We present the defense results of 4 implemented defense strategies.
> >
> > (C.5) Attack performance on different models. We test the impact of 4 more samplers for unconditional attacks.
> >
> > (C.6) Attack performance on SD v2.0. We evaluate the performance on Stable Diffusion v2.0 for Text-to-Image (T2I) attacks.
> >
> > (C.7) Analysis of visualization results. We present the visualization results of the three implemented tools.
> >
> > (C.8) The advantages of MLLM for backdoor evaluation. We summarize the advantages of using MLLM for diffusion backdoor evaluation.
>
> ---
>
> ### **Cons2: Suggestion for highlighting the new conclusions.**
>
> **R2:** Thanks for the valuable suggestion. As the first benchmarks integrating diverse attack types, we have highlighted several new conclusions in boldface at the end of each experiment, for example, the analysis of each backdoor target in Section 4.2 and the analysis of each visualization tool in Appendix C.7. For a better overview, we will summarize some new conclusions to the Introduction Section in the revised version. Here is a brief version:
>
> > Our extensive experiments yield several important findings, encompassing the results of current attacks, defenses, and the visualizations of backdoored models.
> > - **Attacks**: 1) Among the existing backdoor targets, ImageFix and StyleAdd achieve higher ASRs, indicating their greater risk in practical applications. 2) The ObjectRep target attracts the most research attention; however, current attack methods fail to perform consistently across various evaluation metrics. 3) Non-conflicting backdoor targets may be more vulnerable to attack.
> > - **Defenses**: The current performance of model-level defenses is limited for both unconditional DMs (Elijah) and text-to-image (T2I) DMs (T2IShield and Textual Perturbation), whereas the input-level method TERD demonstrates greater effectiveness.
> > - **Visualizations**: 1) *Assimilation Phenomenon*: The ImageFix target exhibits a more pronounced assimilation pattern, while other target types may not show such behavior, indicating the limitations of using attention maps for backdoor visualization.  2) *Activation Norm*: Certain neurons in backdoored DMs consistently show significant differences in activation norms between clean and poisoned samples, suggesting a potential defense strategy for identifying backdoored neurons. 3) *Pre-Activation Distribution*: A clear bimodal distribution is observed in certain neurons, highlighting the need for further investigation into neuron functionality for backdoor defense.
> >
>
> ---
>
> ### **Cons3: Concern about the reliability of MLLM evaluation.**
>
> **R3:** Thanks for your concern. We would like to kindly clarify that the MLLM evaluation is proposed to assess the performance of the backdoored models rather than being directly used for backdoor detection. Therefore, the mentioned “subtle or pixel-level triggers” are not considered to be a potential limitation, where the backdoor-related information should be provided.
>
> As the first work to introduce MLLM for diffusion backdoor evaluation, we follow a similar paradigm as in the previous T2I tasks [1,2] with our own-designed prompts, and keep the pre-trained classifier, e.g., ViT as in EvilEdit [3], in ObjectRep for comparison to ensure its reliability. Moreover, we have conducted further analysis to validate its reliability in Appendix C.1: 1) Figure 4 illustrates the high Pearson correlation of GPT-4o (0.9687) and Qwen2.5-VL-72B (0.9748) with the ViT classifier, indicating their high consistency with the previous evaluation method. 2) Tables 10 and 11 measure the MSE between the MLLM results and the human-annotated results, illustrating the effectiveness of using MLLM to represent human intuition.
>
> To further alleviate your concern, we add a performance comparison between the proposed MLLM evaluation (GPT-4o) and our own-trained binary classifiers following the settings in BadT2I [4]. Specifically, a ResNet18 and a ResNet50 are trained (both with test accuracy > 90%) to evaluate the ASRs of ImagePatch and ObjectRep, respectively. The results shown in Tab. 1 below illustrate that MLLM evaluation is highly reliable with similar performance as in the traditional classifiers.
>
> **Tab. 1. ASR comparisons between classifiers and MLLM.**
>
> | **Attack** | **ASR-Classifier (%)** | **ASR-MLLM (%)** |
> | --- | --- | --- |
> | Pixel-Backdoor (BadT2I) | 100 | 99.6 |
> | TPA (RickRolling) | 97.1 | 96.8 |
> | EvilEdit | 63.5 | 61.1 |
> | DB (PaaS) | 55.5 | 51.3 |
> | TI (PaaS) | 84.7 | 88.7 |
>
> ---
>
> [1] TIFA: Accurate and Interpretable Text-to-Image Faithfulness Evaluation with Question Answering, ICCV 2023.
>
> [2] T2I-CompBench: A Comprehensive Benchmark for Open-world Compositional Text-to-image Generation, NeurIPS 2023.
>
> [3] EvilEdit: Backdooring Text-to-Image Diffusion Models in One Second, MM 2024.
>
> [4] Text-to-Image Diffusion Models can be Easily Backdoored through Multimodal Data Poisoning, MM 2023.

---

> > ### Author Response · Authors · 2025-08-05
> >
> > Dear Reviewer mCTK:
> >
> > We would like to express our sincere gratitude for your valuable insights and suggestions on our work.
> >
> > We have tried our best to address the concerns and queries you raised during the rebuttal process. We would greatly appreciate knowing whether our responses have effectively resolved your concerns. Your feedback will be essential in further improving the quality of our work.
> >
> > Sincerely,
> >
> > Authors

---

> > ### Author Response · Authors · 2025-08-08
> >
> > Dear Reviewer mCTK:
> >
> > Thanks again for your valuable suggestions. As the end of the discussion period is approaching, we would like to kindly ask again for your concerns about our paper. Are there still any unsolved doubts for you?
> >
> > Your help is greatly appreciated. We are eagerly waiting for your feedback before the end.
> >
> > Sincerely,
> >
> > Authors

---

### Note · Authors · 2025-08-13

Dear Program Chairs, Area Chairs, and Reviewers,

We sincerely thank all reviewers for their constructive feedback, which has significantly strengthened our submission. During the rebuttal period, we rigorously addressed every technical concern and further verified our contributions:

**Enhanced Evaluation & Validation**

- We added **classifier comparisons** (ResNet18/50) to validate MLLM reliability (Tab. 1 to Reviewer mCTK), further strengthening Appendix C.1’s conclusion.
- We demonstrated **MLLM consistency** across three attacks and two MLLMs (most ACC/ASR/PSR range ≤1%; Tab. 1 to Reviewers aDMm/Yt6A).
- We implemented **one new SOTA defense** (Tab. 2 to Reviewers aDMm/wD1L), showing consistent limitations that align with our original conclusions.

**Restructured Content & Clarity**

- We summarized and integrated **key findings** (attack/defense/visualization insights) into the Introduction Section.
- We added **Figure 2 interpretation** to Section 3.1 and **target-type examples** to Section 3.2.
- We consolidated the **Appendix summaries** in Section 4.3.
- We **proofread** the full manuscript and **corrected all typos** (e.g., "Input-level"→"Input level").

**Extended Reproducibility Proof & Robustness Analysis**

- We added step-by-step **reproducibility** scripts for T2IShield detection metrics (Precision/Recall/F1), matching original paper results (Tab. 1 to Reviewer wD1L).
- We tested defense robustness under **textual perturbations** (synonym replacement), revealing vulnerability trends (Tab. 2 to Reviewer Yt6A).

Following these substantial revisions, we believe *BackdoorDM* now offers a more rigorous, reproducible foundation for backdoor research on diffusion models, and we look forward to sharing the improved version with the community.

Thank you once again for your effort in facilitating this process.

Sincerely,

Authors

---

### Decision · Program_Chairs · 2025-09-18

**Decision:**

Accept (poster)

**Comment:**

This paper introduces **BackdoorDM**, the first comprehensive benchmark for backdoor learning in diffusion models (DMs), addressing a critical gap in evaluating generative AI vulnerabilities. The work establishes a unified framework for classifying backdoor attacks/targets, introduces a novel multimodal large language model (MLLM)-based evaluation method, and provides a standardized comparison of 9 attack methods, 4 defenses, and 3 visualization tools.

### Key Contributions
1. **Unified Benchmark**:
   - Systematizes fragmented evaluation protocols across unconditional DMs, text-to-image (T2I) pipelines, and diverse attack/target types (e.g., ImageFix, StyleAdd).
   - Proposes a standardized taxonomy and metrics (ASR, PSR, ACC) for fair comparisons.

2. **MLLM-Driven Evaluation**:
   - Pioneers GPT-4o/Qwen2-VL for assessing attack success rate (ASR), perturbation sensitivity (PSR), and clean accuracy (ACC).
   - Validates MLLM reliability via correlation with classifiers (Pearson *r* > 0.96) and multi-seed consistency (±1% variance).

3. **Empirical Insights**:
   - Highlights high-risk attack targets (e.g., ImageFix achieves 99.6% ASR) and defense limitations (input-level TERD outperforms model-level methods).
   - Reveals DM-specific backdoor artifacts (e.g., bimodal neuron activation distributions, assimilation phenomena in ImageFix).

Despite initial concerns about novelty and scope, the paper delivers a technically sound, impactful benchmark that unifies evaluation standards, exposes critical vulnerabilities, and guides future research. The authors’ responsiveness and transparency—evidenced by extensive revisions and supplementary materials—strengthen the paper’s merit. This work is a cornerstone for trustworthy AIGC and warrants acceptance.